# It Takes Two: On the Seamlessness between Reward and Policy Model in RLHF

## Abstract

Reinforcement Learning from Human Feedback (RLHF) involves training policy models (PMs) and reward models (RMs) to align language models with human preferences. Instead of focusing solely on PMs and RMs independently, we propose to examine their interactions during fine-tuning, introducing the concept of **seamlessness**. Our study starts with observing the saturation phenomenon, where continual improvements in RM and PM do not translate into RLHF progress. Our analysis shows that RMs fail to assign proper scores to PM responses, resulting in a 35% mismatch rate with human preferences, highlighting a significant discrepancy between PM and RM. To measure seamlessness between PM and RM without human effort, we propose an automatic metric, SEAM. SEAM quantifies the discrepancies between PM and RM judgments induced by data samples. We validate the effectiveness of SEAM in data selection and model augmentation. Our experiments demonstrate that (1) using SEAM-filtered data for RL training improves RLHF performance by 4.5%, and (2) SEAM-guided model augmentation results in a 4% performance improvement over standard augmentation methods.

## 1 Introduction

Reinforcement learning from human feedback (RLHF) has emerged as a popular technique to optimize and align a language model with human preferences (Stiennon et al., 2020; Nakano et al., 2021; Menick et al., 2022; Glaese et al., 2022; Ouyang et al., 2022; Touvron et al., 2023; Achiam et al., 2023; Bai et al., 2023; Rafailov et al., 2024). RLHF provides a natural solution for optimizing non-differentiable, scalar objectives for language models and has been the centerpiece of recent state-of-the-art large language models (LLMs) (Lu et al., 2022; Hejna III & Sadigh, 2023; Go et al., 2023; Korbak et al., 2023; Achiam et al., 2023; OpenAI, 2023). In RLHF, a *reward model* (RM) generates scalar rewards for a *policy model* (PM) generated outputs as supervision signals during reinforcement learning. Since policy gradient methods (Schulman et al., 2017) optimize based on such signal, the PM and RM inevitably dictate the behavior of the resultant RLHF model. As such, the properties of RMs (or PMs) and their impact on RLHF models have become points of interest for the community (Gao et al., 2023; Zhu et al., 2023; Dong et al., 2023; Gao et al., 2023; Shen et al., 2023). Unlike prior work that examines the individual capabilities of each model, in this work, we introduce and explore the concept of *seamlessness* between the PM and RM, focusing on their interactions.

Our study begins with the observation of a *saturation phenomenon* in the RLHF process (§3): beyond a certain threshold, improvements in the quality of the RM and PM do not translate into increased RLHF performance (Figure 1). To understand this phenomenon, we explore whether the RM can assign appropriate scalar rewards to responses $r$ generated by the PM prompted by instruction $I$. This inquiry addresses the **seamlessness** between the RM and PM. Although the RM performs well on standard preference benchmarks, it struggles to evaluate PM-generated responses effectively. This is demonstrated by a 35% mismatch rate between reward scores and human preferences, indicating a significant, persistent discrepancy between the RM and PM as reflected in the reinforcement learning (RL) training data. This discrepancy does not diminish even as the PM and RM are individually optimized according to their respective evaluation paradigms, thus disrupting their seamlessness. Remarkably, when we remove instructions from the RL dataset that contribute to this discrepancy and re-conduct RLHF, we observe an improvement in RLHF performance. This outcome suggests that enhancing the seamlessness between PM and RM benefits the overall RLHF process.

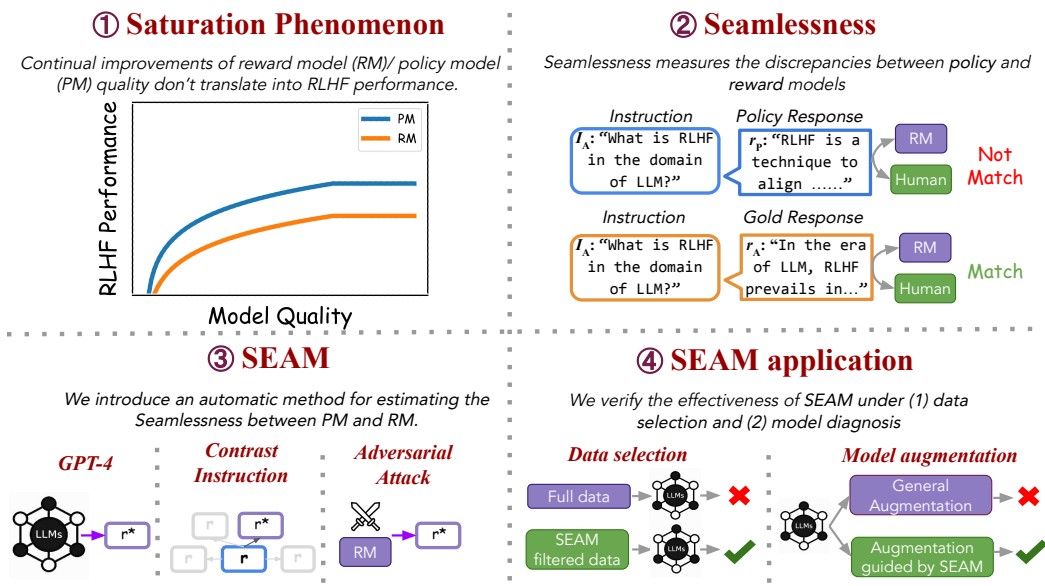

Figure 1: We introduce the concept of *Seamlessness* to measure the *discrepancies* between reward and policy models as supported by human evaluation. To automate measuring the *Seamlessness*, we propose SEAM, an automated method for estimating seamlessness between PM and RM. We validate its effectiveness through two experimental settings: data selection and augmentation.

Based on these findings, we define the seamlessness between the PM and RM as detailed in §5 and introduce an automated estimation method, SEAM, available in three computational variants: $SEAM_{Adv}$, $SEAM_{Contrast}$, and $SEAM_{GPT}$. Such methods remove the reliance on manual effort traditionally required for measuring seamlessness. Essentially, SEAM evaluates the risk associated with each data sample when employed in RLHF processes, considering the specifics of the given PM and RM. Additionally, we give two experimental scenarios to demonstrate how SEAM can be effectively utilized to improve the real-world RLHF process. (1) Data Selection: We compute the SEAM score for each sample and exclude those with low scores for RL training data selection. This strategy underscores a "less is more" phenomenon (Zhou et al., 2024), whereby RLHF performance is enhanced when using this filtered dataset compared to the unfiltered dataset. Additionally, removing low-score samples helps mitigate the "saturation phenomenon". (2) Model augmentation: During RLHF, we explore the PM and RM failure modes and subsequently strengthen them based on identified weaknesses. We calculate the SEAM score for each data sample throughout the RL training phase. Samples exhibiting low SEAM scores are then selected as targets for data augmentation to enhance the capabilities of the PM and RM specifically for these challenging samples. The results show that the SEAM score effectively functions as a diagnostic metric within the RLHF framework. The primary contributions of this paper are three-fold:

- We shift focus from the individual capacities of the reward model (RM) and policy model (PM) to explore their interplay and a noted saturation phenomenon in RM/PM quality. Our analysis identifies a discrepancy between RM and PM that cannot be resolved merely by scaling up.

- We conceptualize the seamlessness between PM and RM and introduce SEAM, an automatic estimation method that quantifies the seamlessness between PM and RM in a data-centric manner.

- We empirically design two experimental scenarios to demonstrate how SEAM can be leveraged to improve RLHF training: (1) Data selection and (2) Model augmentation. Our results validate the effectiveness of SEAM under such scenarios.

## 2 RELATED WORK AND BACKGROUND

**RLHF in Language Models.** In earlier studies, reinforcement learning (RL) has been applied across various domains, such as machine translation (Sokolov et al., 2016; Kreutzer et al., 2018; Nguyen et al., 2017), dialogue generation (Li et al., 2016; Yi et al., 2019; Keneshloo et al., 2019), and text

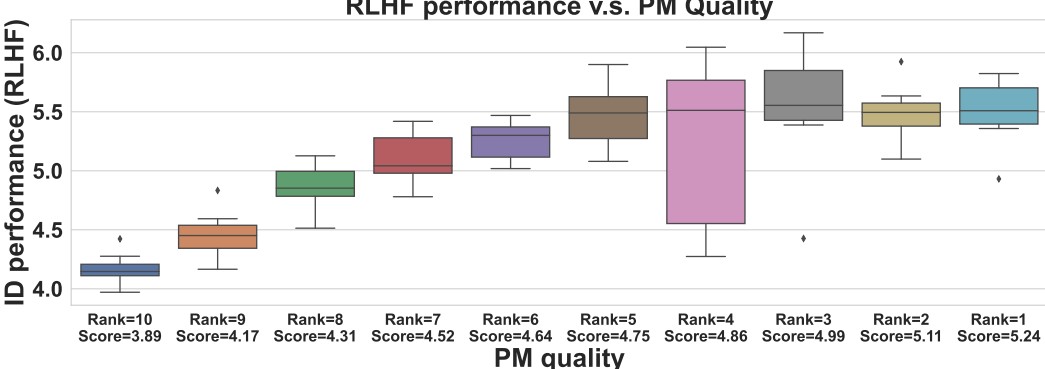

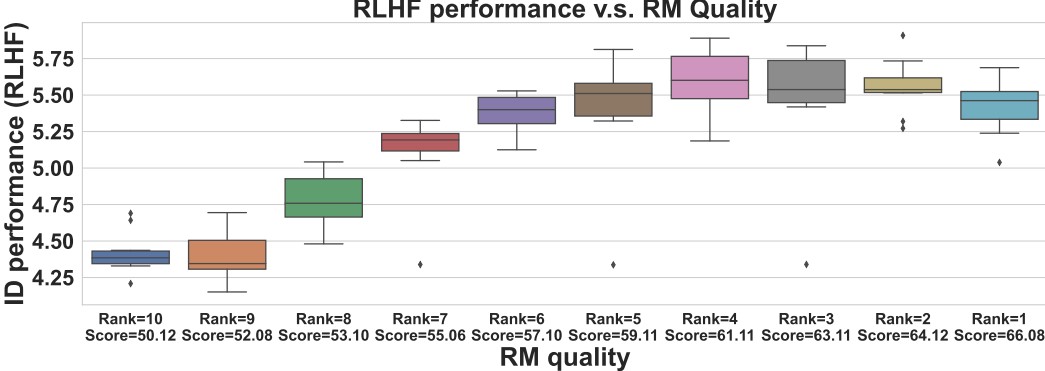

Figure 2: We examine the relation between the RLHF performance and the quality of PMs and RMs, measured by $\mathcal{Q}_{PM}$ and $\mathcal{Q}_{RM}$, respectively. We can see a "saturation phenomeno": the continual improvements of RM/PM do not translate into RLHF improvements.

generation (Li et al., 2018; Ziegler et al., 2019; Shi et al., 2018; Stiennon et al., 2020), often employing modeling reward as automatic evaluation metrics like BLEU (Papineni et al., 2002) or using simulated feedback (Nguyen et al., 2017; Keneshloo et al., 2019). While integrating RL and language models has been extensively explored, significant advancements in RLHF with LLMs for general language tasks have only recently emerged (Ouyang et al., 2022; Touvron et al., 2023; Achiam et al., 2023; Bai et al., 2023; Rafailov et al., 2024). In RLHF, human feedback is collected to train a reward model, which then serves as a surrogate for human feedback during the training process, providing scalar evaluative feedback to the policy model (see detailed background of RLHF in Appendix A). In RLHF, RL algorithms (e.g., PPO (Schulman et al., 2017)) are particularly suitable for training PM and RM.

**Reward Hacking.** In RLHF, a critical issue closely related to our research is "reward hacking", as identified in prior studies (Askell et al., 2021; Pan et al., 2021; Skalse et al., 2022; Shen et al., 2023). This phenomenon arises from **discrepancies between the reward model (RM) and actual human preferences** (Gao et al., 2023; Lambert & Calandra, 2023). Although optimizing towards maximizing the rewards may initially appear beneficial, it ultimately leads the trained policy to exploit loopholes in the RM, securing high rewards without achieving the intended objectives. This degrades performance, complicates the selection of effective checkpoints, and may produce outputs that do not genuinely reflect human preferences (Singhal et al., 2023). Such misalignments increase tendencies towards sycophancy (Perez et al., 2023), reinforcing social biases (Santurkar et al., 2023; Ziems et al., 2024) and pose safety risks (Ngo et al., 2022; Carlini et al., 2024; Shen et al., 2024). A key distinction of our work is its focus on **the discrepancies between RM and PM**, which we term 'seamlessness', as opposed to the traditional focus on discrepancies between reward models and human values.

## 3 THE SATURATION PHENOMENON REFLECTED IN RLHF QUALITY

In this section, we conduct experiments to investigate the relationship between the RLHF outcomes and the quality of PM/RM.

**Experimental Setup.** We follow the experimental configuration of StackLLaMa (Beeching et al., 2023) due to the proven success of its PPO and data settings for RLHF. Our framework employs the LLaMa2-7B model as the base model for both the reward and policy models. To explore the

effects of the quality of RM and PM, we change the volume of training data, enabling us to produce a spectrum of model strengths for both PM and RM. We develop ten variants each for RMs and PMs. Each pairing of PM and RM is then subjected to the RLHF technique, resulting in hundreds of unique RLHF models. Further details on implementation and setup are provided in Appendix C.

**Quality Metrics.** We employ two metrics[1] to assess the quality of the PM and RM: $\mathcal{Q}_{PM}$ (PM quality) and $\mathcal{Q}_{RM}$ (RM quality). In our experiments on StackExchange, $\mathcal{Q}_{PM}$ measures how well the policy model generates answers to StackExchange questions. We use 1000 samples from the StackExchange test split, with responses generated by the LLM evaluated by GPT-4 on a scale from 1 (worst) to 10 (best), similar to the MT-Bench scale. On the other hand, $\mathcal{Q}_{RM}$ evaluates the accuracy of the reward model in predicting human preferences on the StackExchange preference benchmark test split. Additional details are provided in Appendix C.

**Results.** We show the correlation between the in-domain performance ($\mathcal{Q}_{PM}$ and $\mathcal{Q}_{RM}$) of RLHF models and the quality of RMs and PMs, as illustrated in Figure 2. Our primary observation is that while the quality of RMs and PMs generally positively correlates with the in-domain performance of RLHF models, a *saturation effect* is evident. Beyond a certain quality threshold, additional RM or PM quality improvements yield no further enhancements in the in-domain performance of RLHF models.

## 4    ANALYZING THE ORIGIN OF SATURATION PHENOMENON

This section investigates the saturation phenomenon within RLHF, particularly from the perspective of potentially noisy supervision signals. The RLHF process comprises three primary stages: (1) policy modeling, (2) reward modeling, and (3) RL training. Initially, we conduct a sanity check on our PM and RM under our experimental settings to confirm their capacity for transferability across different data subsets. As shown in §4.1, the results indicate that both the PM and the RM exhibit adequate generalization. During the RL training stage, however, we observe that the RM struggles to effectively evaluate many of the responses generated by the PM (§4.2). By removing data that reflects this discrepancy between RM and PM, we find that RLHF performance improves (§4.3).

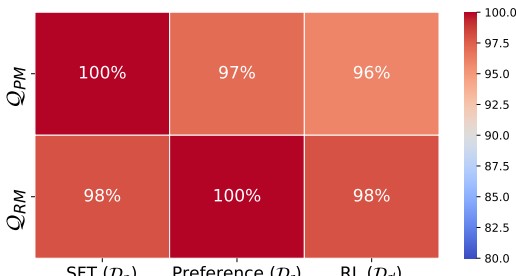

Figure 3: Cross-validation of PM and RM quality using different datasets(3 random seeds). The performance of RM and PM remains consistent across benchmarks. (e.g., on $\mathcal{D}rl$, the PM achieves 96% of its performance on $\mathcal{D}_p$.)

### 4.1    A SANITY CHECK ON PM AND RM

We hypothesize that the observed saturation phenomenon may be due to the capacity of RM or PM can not be transferred to data used in other stages (e.g., the policy model can generate high-quality responses towards SFT instructions but fails to respond to the RL instructions). Thus, we conducted a sanity check on both models to answer the following two questions: (1) Q1: whether the RM consistently distinguishes between better and worse responses as per the instructions used in SFT and RL training and (2) Q2: whether the PM sustains its generation quality with instructions from the RL dataset. We prepare the SFT dataset $\mathcal{D}_p$, the preference benchmark $\mathcal{D}_r$, and the RL dataset $\mathcal{D}_{rl}$. Specifically, the PM and RM were trained on the train splits of $\mathcal{D}_p$ and $\mathcal{D}_r$, respectively. We then employed cross-validation techniques to assess the PM's performance across the test split of the preference and RL datasets. Similarly, we tested the RM on the test split of the SFT and RL datasets. Experimental details are deferred to Appendix C.

The results are shown in Figure 3. We trained five models each for the PM and RM, subsequently performing cross-validation. The key observation is that the performance of both PM and RM remains consistent across various in-domain datasets. This consistency indicates that PM and RM do not have significant generalization issues under our experimental setup. Besides, it also answers our two questions: (1) Given a well-trained PM that performs well on the evaluation set of $\mathcal{D}_p$, it can also

---

[1]We do not use the KL divergence between the outputs from the reference and policy models, as there is no clear correlation between model quality and such KL divergence.

respond with similar quality to the instructions in $\mathcal{D}_{rl}$; (2) Given a well-trained RM that performs well on the evaluation set of $\mathcal{D}_r$, it can also perform similarly well on distinguishing the golden and worse response in $\mathcal{D}_p$ and $\mathcal{D}_{rl}$.

## 4.2 DISCREPANCY BETWEEN RM AND PM DURING RL TRAINING

During the RL training stage, the PM is prompted by instructions from the RL dataset $\mathcal{D}_{rl}$ to generate responses $r_i$. The RM then evaluates these responses, which assigns reward scores to guide the RL training process. Our empirical analysis reveals two key findings (Figure 3), given a high-quality PM and RM: (1) the RM can effectively discriminate between golden and suboptimal responses of instructions within $\mathcal{D}_{rl}$, and (2) the PM can generate high-quality responses to instructions from $\mathcal{D}_{rl}$. Thus, we investigate the RM's capacity to evaluate the PM's responses to $\mathcal{D}_{rl}$ since there might be a distribution shift between the responses generated from PM and those in the dataset.

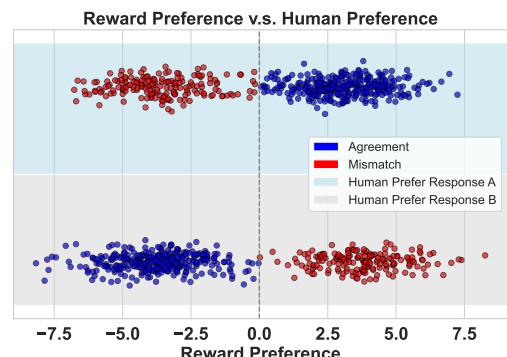

Figure 4: Agreement between reward and human preference is evaluated by comparing two responses (A and B) from two different policy models. The blue points indicate agreement between the reward and human preferences, while the red points represent mismatches. However, the results show that the RM fails to assign a proper score to the generation from PM.

Directly evaluating the RM capability to accurately assign scores to responses generated by the PM conditioned on an instruction $I_i$ has significant challenges since the standard reward modeling cast the preference regression problem into a classification problem. To address this, we employ a comparative analysis. We select two PMs of differing qualities (ranked 1 and 5 in previous experiments §3) and prompt each PM with instructions from the dataset $\mathcal{D}_{rl}$ (we sample a total of 1,000 instructions). We collect the responses and organize them into pairs for evaluation. Each pair of responses is evaluated by two methods: (1) human judgment and (2) RM evaluation using the rank 1 RM from §3 to determine if even our best RM faces issues. To investigate the matching degree between RM and human preferences, we present pairs of responses (A and B) from the two PMs to human annotators without revealing the originating model. Human annotators are asked to annotate their preference between the two options. Similarly, we determine RM preferences based on their assigned reward scores.

The results, as shown in Figure 4, reveal a mismatch rate of approximately $40\%$, showing that the RM has some inability to accurately assign scores that reflect the true quality of responses generated by the PM. Also, we can observe a discrepancy between PM and RM - the RM can not well judge the quality of the responses generated from PM. This discrepancy can introduce noise into the RL training process, leading to the accumulation of incorrect gradients during RL optimization. Besides, we show that such discrepancies can not be resolved by scaling up the model (Appendix B). Consequently, a natural strategy to enhance the RLHF process is removing instructions from $\mathcal{D}_{rl}$ that exhibit discrepancies between the RM and PM. This approach aims to reduce the noise in the RL training procedure, potentially improving overall model performance.

## 4.3 LESS CAN BE MORE: A CASE STUDY OF DATA SELECTION FOR RL TRAINING

Based on the insights from §4.2, we remove instructions that lead to discrepancies between the PM and RM. We then use this refined dataset for RL training and compare its performance against that achieved using the full $\mathcal{D}_{rl}$ dataset. As per the experimental settings described in §4.2, we employ both models at rank= 1 for RL training. The results, presented in Figure 5, demonstrate a statistically significant improvement in RLHF performance (p<0.05) after removing data that causes discrepancies between the PM and RM. This case study illustrates a 'less is more' phenomenon in RL training data: removing data that causes the discrepancy between PM and RM can enhance overall RLHF performance. However, this selective data filtering process is challenging to generalize due to its dependence on human annotation. Currently, there is no formal concept to characterize such data-driven discrepancies adequately. Consequently, we will discuss these in §5.

# 5 SEAM: AN AUTOMATIC ESTIMATION FOR SEAMLESSNESS

As shown in §4, removing data that leads to discrepancies between the PM and the RM improves RLHF performance. Currently, our approach depends on manual human assessments to determine the alignment between the PM and RM for specific datasets, a process that hinders full automation. This section first explores the concept of 'seamlessness' in RL training data. Then, we propose SEAM, an automated method designed to quantify the seamlessness of each data point, potentially enabling a more efficient and systematic tool to enhance RLHF training.

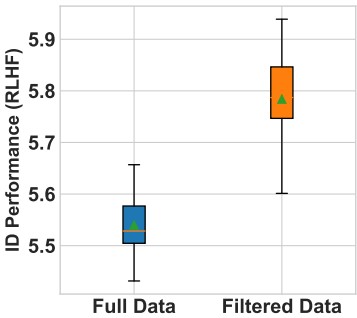

Figure 5: Compared to the RLHF performance of the full dataset, filter low-SEAM data further improves RLHF (3 random seeds).

## 5.1 CONCEPT OF THE SEAMLESSNESS

Generally, our concept of 'seamlessness' is proportional to the PM likelihood of a data point that causes discrepancies between the policy and the reward model. Therefore, seamlessness includes not only the probability of misjudgment by the reward model but also the generative distribution of the policy model when conditioned on given data. The formal definition of seamlessness is provided in Definition 1. Considering that it is implausible to iterate the space of all responses $r$, we provide a discretization form for seamlessness in Equation 2.

---

**Definition 1.** (**Definition of Seamlessness**) Given an instruction $I \in \mathcal{D}_{rl}$, a reward model $\mathcal{R}_\theta$ and a policy model $\pi^{\text{SFT}}$. We denote the distribution of the response $r$ from $\pi^{\text{SFT}}$ as $P_r(\cdot|I, \pi^{\text{SFT}})$, we also denote the data distribution that hacks $\mathcal{R}_\theta$ as $P_h(\cdot|\mathcal{R}_\theta)$, which means the data that leads to reward misjudgement. Then, the seamlessness of the instruction $I$ is defined as follows:

$$\mathcal{S}(I, R_\theta, \pi^{\text{SFT}}) = \int_{r \sim P_h} P_r\left(r \mid I, \pi^{SFT}\right) \cdot \epsilon(r, R_\theta)\, dP_h \tag{1}$$

where $\epsilon(r, R_\theta)$ denotes the magnitude of RM misjudgement.

---

Since the term defined in Definition 1 is intractable, we propose SEAM, an estimation for the seamlessness between RM and PM reflected through data. Following the notations in Definition 1, we define a sample set $\mathcal{X}$ that contains $N$ samples $r_i \sim P_h(\cdot|\mathcal{R}_\theta)$ to represent the hacking distribution. Then, we present the discretization form of the seamlessness as follows:

$$\text{SEAM}(I, R_\theta, \pi^{\text{SFT}}) = \sum_{r_i \in \mathcal{X}} P_r\left(r_i \mid I, \pi^{SFT}\right) \cdot \epsilon(r_i, R_\theta) \tag{2}$$

In fact, our analyses in §4 use a similar method to Equation 2 to quantify the seamlessness between PM and RM. But under the formulation in §4, the $\epsilon(r_i, R_\theta)$ refers to the mismatch degree between reward and human preferences, which inevitably incorporate the human efforts.

## 5.2 AUTOMATIC ESTIMATION FOR SEAMLESSNESS

A significant practical challenge in our previous method of measuring seamlessness is the difficulty in automating the process. In this part, we introduce several automated estimation methods designed to quantify the seamlessness of data. Specifically, we propose three variants based on their corresponding designs to construct the sample set $\mathcal{X}$ (Equation 2): SEAM_Contrast, SEAM_GPT, SEAM_Adv.

**SEAM_Contrast** In the SEAM_Contrast method, we implement the 'Contrast Instruction' strategy (Shen et al., 2023) to automatically construct the sample set $\mathcal{X}$. Specifically, for each instruction and its golden response pair $(I, r)$ in the dataset $\mathcal{D}_{rl}$, we retrieve 30 semantically relevant but distinct instructions $I^*$, along with their corresponding golden responses $r^*$, from a large SFT dataset (each pair in this dataset comprises an instruction and its golden response). We then use $r^*$ to form new pairs, assessing whether the reward model can effectively distinguish between the quality of the original pair $I \circ r$ and the newly constructed pair $I \circ r^*$. It is guaranteed that the quality of $I \circ r$ is superior to $I \circ r^*$, providing a reliable ground truth for evaluating RM performance. We define the

magnitude of RM misjudgments, $\epsilon(r_i, R_\theta)$, as follows:

$$\epsilon(r_i, R_\theta) = \max\left\{R_\theta(I \circ r^*) - R_\theta(I \circ r), 0\right\} \tag{3}$$

**SEAM$_{\mathbf{GPT}}$**  In the SEAM$_{\text{GPT}}$ method, we use GPT-4 (Achiam et al., 2023) to construct the sample set $\mathcal{X}$. Specifically, for each instruction and its golden response pair $(I, r)$ in the dataset $\mathcal{D}_{rl}$, we prompt GPT-4 to produce worse-quality responses $r^*$. Similarly, we use $r^*$ to form new pairs, assessing whether the reward model can effectively distinguish between the quality of the original pair $I \circ r$ and the newly constructed pair $I \circ r^*$. We reuse the magnitude defined in Equation 3.

**SEAM$_{\mathbf{Adv}}$**  In the SEAM$_{\text{Adv}}$ method, we use the adversarial attack to generate adversarial sentences that construct the sample set $\mathcal{X}$. Specifically, for each instruction and its golden response pair $(I, r)$ in the dataset $\mathcal{D}_{rl}$, we use adversarial attacks (Ren et al., 2019) to produce responses $r^*$ that hacks the reward model, such that $R_\theta(I \circ r^*) > R_\theta(I \circ r)$. Similarly, we follow the misjudgment term defined in Equation 3.

**Length penalty term**  We introduce the operation to remove length bias. This operation targets the bias introduced by the length of response $r$, primarily affected by the exponential decrease in probability with increasing sequence length. To mitigate this, we implement a length normalization operation on the log probability of the response. This is formally represented as $\frac{\log P_r\left(r_i | I, \pi^{\text{SFT}}\right)}{\text{len}(r_i)}$, where $\log P_r(r_i \mid I, \pi^{\text{SFT}})$ denotes the logarithm of the probability that the policy model assigns to generating the response $r_i$ given the instruction $I$.

## 6  THEORETICAL ANALYSIS

The SEAM method aims to enhance model performance by filtering out instructions that lead to responses difficult for the reward model (RM) to evaluate reliably. We analyze how filtering instrcutions can improve the performance of the policy model obtained through reinforcement learning with human feedback (RLHF).

Let the parameter space $\Pi$ contain the ground-truth distribution $\mathbb{P}(r|I) = P_r(r|I; \pi^{\text{DATA}})$ for some $\pi^{\text{DATA}} \in \Pi$. For simplicity, we assume that all the distributions in this section have common support. We model RLHF as a two-step optimization process: (1) RM optimization and (2) PM optimization.

**Step 1. RM Optimization**  Let $Q_I(\cdot)$ be the distribution of instructions. Then, the RM is obtained by minimizing the following loss (Ouyang et al., 2022):

$$-\mathbb{E}_{\substack{I \sim Q_I(\cdot) \\ r \sim P_r(\cdot|I; \pi^{\text{DATA}}) \\ r' \sim P_r(\cdot|I; \pi^{\text{SFT}})}}\left[\log \sigma\left(R(I \circ r) - R(I \circ r')\right)\right], \tag{4}$$

where $\sigma$ is the sigmoid function. Assume that the class of RMs $\mathcal{R}$ contains $R(I, r) = Z(I) + \log\{P_r(r|I; \pi^{\text{DATA}})/P_r(r|I; \pi^{\text{SFT}})\}$, where $Z(I)$ is any function of $I$. Let $R^* \in \mathcal{R}$ be the minimizer of the loss in equation 4.

**Step 2. PM Optimization**  We consider the following reward maximization problem (Jaques et al., 2017; 2020; Rafailov et al., 2024):

$$\max_{\pi \in \Pi} \mathbb{E}_{\substack{I \sim Q_I(\cdot) \\ r \sim P_r(\cdot|I; \pi^{\text{DATA}})}}[R^*(I, r)] - \beta \mathbb{E}_{I \sim Q_I(\cdot)} D_{\text{KL}}(P_r(\cdot|I; \pi) \| P_r(\cdot|I; \pi^{\text{SFT}})), \tag{5}$$

where $\beta > 0$. The solution to this maximization problem is denoted by $\pi^{\text{RLHF}} \in \Pi$.

### 6.1  INSTRUCTION FILTERING VIA SEAM

SEAM selectively removes instructions that yield responses difficult to evaluate by the RM. Since this corresponds to filtering out samples where $P_r(r|I; \pi^{\text{SFT}})$ is large, while $P_r(r|I; \pi^{\text{DATA}})$ is small, the SEAM method can be seen as filtering out samples where the KL divergence between the responses generated by $\pi^{\text{DATA}}$ and $\pi^{\text{SFT}}$ is large. For the economy of notation, for any divergence $D$, we denote $D(\pi\|\pi'|I)$ as $D(P_r(\cdot|I; \pi)\|P_r(\cdot|I; \pi'))$.

For $\eta > 0$, let $\mathcal{I}_\eta$ represent the set of "good" instructions, where the KL divergence between the responses generated by $\pi^{\text{DATA}}$ and $\pi^{\text{SFT}}$ is small: $\mathcal{I}_\eta = \{I : D_{\text{KL}}(\pi^{\text{DATA}} \| \pi^{\text{SFT}} | I) \leq \eta\}$. Suppose that we could train a policy model that mirrors $\pi^{\text{RLHF}}$ on good instructions and $\pi^{\text{SFT}}$ otherwise. In other words, assume there exists some $\tilde{\pi}^{\text{RLHF}} \in \Pi$ such that

$$
P_r(r|I; \tilde{\pi}^{\text{RLHF}}) = \begin{cases} P_r(r|I; \pi^{\text{RLHF}}) & \text{if } I \in \mathcal{I}_\eta, \\ P_r(r|I; \pi^{\text{SFT}}) & \text{otherwise.} \end{cases}
$$

We note that in contrast to our practical approach, the theory assumes post-filtering of instructions after training on the full dataset, which simplifies the understanding of model behavior by deriving the effect of 'bad' instructions on the alignment between the policy model (PM) and reward model (RM).

Let $Q_I'(\cdot)$ be any distribution of instructions to test the performance of the PMs. Denote $D_\alpha$ as the Rényi divergence of order $\alpha$.

**Proposition 1.** *For any $\beta \in (0, 1)$ and $\eta > 0$, if $\mathbb{P}_{I \sim Q_I'(\cdot)}(I \notin \mathcal{I}_\eta) > 0$, $\max_I D_\infty(\pi^{\text{DATA}} \| \pi^{\text{SFT}} | I) < \infty$, and $\min_{I \notin \mathcal{I}_\eta} D_{\text{KL}}(\pi^{\text{DATA}} \| \pi^{\text{SFT}} | I) > 0$ hold, then,*

$$
\liminf_{\beta \downarrow 0} \frac{\mathbb{E}_{I \sim Q_I'(\cdot)}\left[D_{\text{KL}}(\pi^{\text{DATA}} \| \pi^{\text{RLHF}} | I)\right] - \mathbb{E}_{I \sim Q_I'(\cdot)}\left[D_{\text{KL}}(\pi^{\text{DATA}} \| \tilde{\pi}^{\text{RLHF}} | I)\right]}{\mathbb{E}_{I \sim Q_I'(\cdot)}[D_{\text{KL}}(\pi^{\text{DATA}} \| \pi^{\text{RLHF}} | I)]} > 0.
$$

Proposition 1 shows that the RLHF model with instruction filtering performs strictly better than the RLHF model without instruction filtering when (1) $\beta$ is sufficiently small, and (2) PM with $\tilde{\pi}^{\text{SFT}}$ is uniformly different from data generating process $\pi^{\text{DATA}}$. More details are deferred to Section F in the appendix.

## 7  SEAM FOR RL TRAINING DATA SELECTION

In this section, we employ three SEAM variants as indicators to filter RL training data and evaluate the corresponding effectiveness.

### 7.1  EXPERIMENTAL SETUP

Since this is a data-centric experiment, we follow the previous RLHF setup outlined in Appendix C. For SEAM$_{\text{Contrast}}$, we utilize SimCSE (Gao et al., 2021) as the embedding model to retrieve the top 30 instructions from a StackExchange dataset containing over 1 million instruction-response pairs, with cosine similarity values in the interval $[0.8, 0.9]$. For SEAM$_{\text{GPT}}$, we select GPT-4-0613 to generate 30 lower-quality responses using the prompt shown in Prompt 2. For SEAM$_{\text{Adv}}$, we employ TextAttack (Morris et al., 2020) to perform adversarial attacks on the reward model. For each instruction, we generate 30 adversarial responses.

For the models, we reuse the policy model and reward model checkpoints from §3 to calculate each SEAM variant across the RL dataset. Subsequently, we filter out 20% of the RL dataset based on the value of each SEAM variant, respectively. We then compare the RLHF performance using the full and filtered datasets based on the evaluation paradigm used in §3. Specifically, we add a baseline (**LLaMa**) that uses the perplexity computed by LLaMa2-7B and filters the high perplexity data.

### 7.2  RESULTS

The results are presented in Figure 6, showcasing performance based on the top-5 RMs and PMs, where the saturation phenomenon occurs (§3). The key observations are as follows:

(1) Training on SEAM-filtered RL data further improves RLHF performance: Compared to RLHF on the full $\mathcal{D}_{rl}$, conducting RL training on the filtered $\mathcal{D}_{rl}$ enhances RLHF performance. This finding empirically validates that data with low SEAM values negatively impacts the RL training stage in RLHF. Additionally, randomly removing the same amount of RL training data does not yield benefits, indicating that the effectiveness of SEAM is not merely due to a reduction in data size.

(2) Training on SEAM$_{\text{GPT}}$-filtered RL data alleviates the saturation phenomenon: We observe that as the quality of RM (PM) increases, conducting RLHF on the data filtered by SEAM$_{\text{GPT}}$ continues to improve performance to a certain extent. Compared to the case of full data training, the saturation phenomenon is mitigated by filtering data with low SEAM$_{\text{GPT}}$ values.

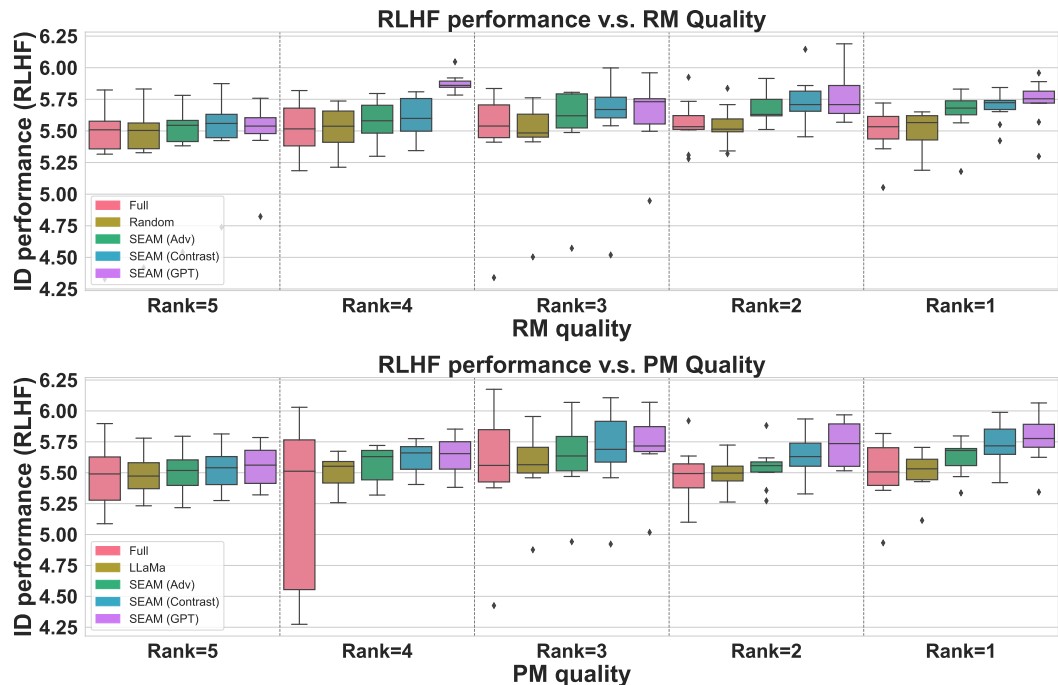

Figure 6: RLHF performance when using SEAM to filter 20% of the RL dataset $\mathcal{D}_{rl}$. After filtering out the low-SEAM data, we observe an improvement in RLHF performance compared to using the full $\mathcal{D}_{rl}$. The effectiveness of the three SEAM variants is ranked as follows: **GPT > Contrast > Adv**. Specifically, we also observe that randomly removing 20% RL data does not bring statistically significant performance changes.

In general, the performance of the three SEAM variants is ranked as follows: **GPT > Contrast > Adv**. In this section, we analyze the limitations of these variants through case studies and a straightforward analysis. Under the setup in Equation 2, a low likelihood indicates that, given the instruction $I$, the PM is unlikely to generate the response $r^* \in \mathcal{X}$, leading to issues in estimating seamlessness.

For SEAM$_{\text{Adv}}$, we found that the adversarial sentences generated for estimating SEAM have a much lower likelihood in the

| SEAM | ATTACK | LIKELIHOOD |
|---|---|---|
| SEAM$_{\text{GPT}}$ | - | -1.81 |
| SEAM$_{\text{Contrast}}$ | - | -3.07 |
| SEAM$_{\text{Adv}}$ | GA | -9.32 |
| | BA | -9.17 |
| | PWWS | -9.87 |

Table 1: Per-sentence log-likelihood (with length penalty) from the top-ranked PM (rank 1) for sentences in the sample set $\mathcal{X}$ (Equation 2) computed using the three estimation variants of SEAM. The sentences created by SEAM$_{\text{Adv}}$ exhibit significantly lower likelihoods, indicating their unnaturalness.

PM compared to the other two methods, as shown in Table 1. Compared to the other two variants, the sentences generated by SEAM$_{\text{Adv}}$ are significantly less likely to be sampled from the PM. Although such adversarial sentences can consistently hack the RM, they do not represent the PM's natural outputs, indicating a lack of representativeness. This is because adversarial attacks tend to introduce non-coherent perturbations to the response $r$, significantly impacting fluency. We present typical cases in Appendix D. For SEAM$_{\text{Contrast}}$, a similar low-likelihood problem exists, although it is less severe than with SEAM$_{\text{Adv}}$.

## 8  SEAM FOR RLHF MODEL AUGMENTATION

This section demonstrates how SEAM can augment models that target to increase seamlessness between PM and RM.

## 8.1 Experimental Setup

We maintain our previous RLHF setup and use the same implementation of SEAM. The key difference between this experiment and the one in §7 is that, after computing SEAM for the RL dataset, we augment the PM and RM by adding the data augmented based on such low-SEAM data points in $\mathcal{D}_{rl}$, rather than filtering them.

For each SEAM variant, we select the lowest 20% of the data based on their SEAM scores and generate augmented data to enhance the RM and PM. Specifically, for each low-SEAM instruction $I_i$ and its corresponding golden response $r_i$, we apply the 'Contrast Instruction' strategy (Shen et al., 2023) to create five augmented data samples for each instruction-response pair. These samples are then added to the training set of the PM. Similarly, we use the same method for the RM to generate five augmented preference data samples for each low-SEAM instruction $I_i$, which are incorporated into the RM's training data. We assess the RLHF performance using the augmented PM and RM. To ensure a fair comparison, we add two baselines: (1) *Random*: we randomly select 20% of $\mathcal{D}_{rl}$ and apply the same augmentation method. The RLHF performance of the PM and RM augmented by both SEAM and random selection is then evaluated. (2) *Full Aug*: For each data sample in $\mathcal{D}_{rl}$, we apply augmentation methods based on all of them, and add the augmented data to train PM and RM.

## 8.2 Results

As shown in Figure 7, the results illustrate the effectiveness of using SEAM to guide model augmentation. Augmenting the PM and RM with data specifically selected by SEAM demonstrates greater benefits than augmentations using randomly selected RL data and achieves comparable performance towards *Full Aug*. This indicates that the RL data chosen by SEAM is closely related to the weaknesses of the RM and PM combination during RLHF. Addressing these specific weaknesses through targeted data augmentation effectively improves the identified issues. Overall, this validates that SEAM can serve as a signal to improve RM and PM in terms of their brittleness during RLHF.

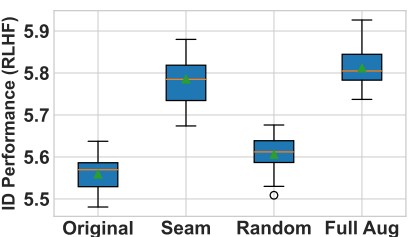

Figure 7: Performance comparison between model augmentation w/ and w/o SEAM. 'Original' means RLHF with no model augmentation.

## 9 Limitations

One limitation of our framework is that it is restricted to offline RLHF experiments rather than being tested in an online RLHF scenario. In online RLHF, the RM and PM are updated continuously based on real-time feedback from user interactions, offering a more dynamic and realistic setting. Despite this, we propose that SEAM can still be effectively utilized by segmenting the online RLHF into a series of offline RLHF cycles. At the beginning of each cycle, the same analyses for data selection and model augmentation could be applied. This adaptation would allow us to extend the benefits of SEAM to more practical, real-world applications. Another limitation of our framework is inherent in the SEAM metric, which assesses the seamlessness of data only comparatively rather than absolutely. Consequently, while we can selectively filter portions of data (e.g., the lowest 20%), we cannot establish a definitive threshold to categorize data as good or bad outright. However, to understand the impact of different filtering rates more thoroughly, we have conducted an analysis detailed in Appendix E, where we see 20% is a practical choice.

## 10 Conclusion

In this paper, we explored the concept of seamlessness between policy and reward models within Reinforcement Learning from Human Feedback (RLHF), uncovering significant discrepancies between the models as reflected in the data. We introduced SEAM, an automated method to quantify this seamlessness, demonstrating its practical benefits for improving RLHF outcomes. Our findings emphasize the critical interplay between policy and reward models, contributing to a deeper understanding of RLHF dynamics. We hope our insights will guide future research toward developing more effective and nuanced RLHF strategies.

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

## A PRELIMINARIES: A THREE-STAGE PARADIGM FOR RLHF

A RLHF practice includes three stages: policy modeling, reward modeling, and RL training, which involve three benchmarks: an SFT dataset $\mathcal{D}_p$, a preference benchmark $\mathcal{D}_r$, and an RL dataset $\mathcal{D}_{rl}$.

**Policy model.** Following the setup (Ouyang et al., 2022), we obtain the policy model (PM) by supervised fine-tuning (SFT) the base version of LLM. Given an SFT dataset $\mathcal{D}_p$, each instance in the dataset consists of an instruction and its golden response. Then, we train the LLM on $\mathcal{D}_p$ with language modeling loss to obtain the PM: $\pi^{\mathrm{SFT}}$.

**Reward model.** Following the conventional setup (Ouyang et al., 2022), we are given a dataset of human preferences $\mathcal{D}_r$. Each instance in this dataset $(I_i, r_i^+, r_i^-)$ is comprised of an instruction prompt $I_i$, a pair of responses $r_i^+, r_i^-$ where $r_i^+$ is preferred over $r_i^-$ by humans. On this labeled data, RM $\mathcal{R}_\theta$ is trained to assign a higher scalar reward to human-preferred $r_i^+$ over non-preferred $r_i^-$ in the context of $I_i$. This can be achieved by minimizing the ranking loss $\mathcal{L}$, where $\sigma$ is the sigmoid function and $I_i \circ r_i^+$ is the concatenation of $I_i$ and $r_i^+$.

$$\mathcal{L}(\theta) = -\mathbb{E}_{(I_i, r_i^+, r_i^-) \sim \mathcal{D}_h} \left[ \log \left( \sigma \left( \mathcal{R}_\theta(I_i \circ r_i^+) - \mathcal{R}_\theta \left( I_i \circ r_i^- \right) \right) \right) \right]. \tag{6}$$

**Reinforcement Learning.** The last stage of RLHF is reinforcement learning. Specifically, a per-token KL penalty from the SFT model at each token is used to mitigate over-optimization of the reward model, and the value function is initialized from the RM. We maximize the following combined objective function $\mathcal{J}(\phi)$ in RL training based on PPO algorithm (Schulman et al., 2017; Ouyang et al., 2022), RL training dataset $\mathcal{D}_{rl}$ and pre-training dataset $\mathcal{D}_{\mathrm{pre}}$:

$$\mathcal{J}(\phi) = \mathbb{E}_{(I,r) \sim \mathcal{D}_{\pi_\phi^{\mathrm{RL}}}} \left[ \mathcal{R}_\theta(I \circ r) - \beta \log \left( \pi_\phi^{\mathrm{RL}}(r \mid I) / \pi^{\mathrm{SFT}}(r \mid I) \right) \right]$$

where $\pi_\phi^{\mathrm{RL}}$ is the learned RL policy parameterized by $\phi$ initialized from a pretrained supervised trained model $\pi^{\mathrm{SFT}}$. The first term encourages the policy $\pi_\phi^{\mathrm{RL}}$ to generate responses that have higher reward scores. The second term represents a per-token KL reward controlled by coefficient $\beta$ between $\pi_\phi^{\mathrm{RL}}$ and $\pi^{\mathrm{SFT}}$ to mitigate over-optimization toward the reward.

## B THE DISCREPANCY DOES NOT VANISH AS SCALING UP

| Model | Match Rate | PM performance | RM performance |
|---|---|---|---|
| LLaMa2-7B | 60.5% | 66.1 | 5.24 |
| LLaMa2-13B | 60.7% | 66.9 | 5.30 |
| LLaMa2-70B | 60.4% | 67.6 | 5.35 |

Table 2: The scaling tendency of our base model for training PM and RM, from 7B to 70B. We observe that the performance of PM and RM improves as the model scales up but find the match rate toward human preference remains nearly the same.

As demonstrated in §4.2, there is a notable discrepancy between the PM and RM: the RM fails to appropriately assign reward scores to responses generated by the PM. In this section, we explore the impact of scaling the base model on these discrepancies by reanalyzing the data discussed in §4.2. The findings, presented in Table 2, reveal that while the capacities of the PM and RM improve with an increase in the size of the base model (LLaMa2), the preference matching rate remains nearly consistent across different model scales. These results confirm that merely scaling up the model size does not address the underlying discrepancy between the RM and PM.

## C IMPLEMENTATION DETAILS OF RLHF

### C.1 TRAINING DETAILS

• Standard fine-tuning (SFT): The base model chosen is LLaMa2-7B. We created 10 PMs of increasing quality by varying the training data amounts at 50, 100, 250, 500, 800, 1500, 2500, 5000, and 10000,

plus a baseline pretrained model without SFT. The configuration employed includes the AdamW (Kingma & Ba, 2014) optimizer with a learning rate of 1e-4, 10 warmup steps, and training facilitated by LoRA.

• Reward model (RM): Training of the RM utilized the SFT model as the base model. Depending on the SFT model's quality rank, StackExchange pairwise preference data of subset 50, 100, 500, 2500, 5000, 10000, 20000, 50000, and 100000 were employed to train 9 RMs. With an additional pretrained model replaced with a randomly initialized classifier head, in total we create 10 RMs with increasing accuracy. Training employed LoRA, with AdamW optimizer and learning rate 2e-5.

• Reinforcement learning with PPO: PPO is used for each PM-RM pairing, generating hundreds of unique RLHF models. The RL prompts are from the StackExchange question dataset and remain consistent across all RLHF implementations. The SFT model served as the reference model, utilizing the reward scores from the RM as supervision. All PPO training has the configuration of LoRA with a learning rate of 1.4e-5, a batch size of 32, and 200 PPO steps.

---

**Prompt 1.** (Prompt used in RLHF/PM evaluation)

[System]

Please act as an impartial judge and evaluate the quality of the response provided by an AI assistant to the user question displayed below. Your evaluation should consider factors such as the helpfulness, relevance, accuracy, depth, creativity, and level of detail of the response. Begin your evaluation by providing a short explanation. Be as objective as possible. After providing your explanation, please rate the response on a scale of 1 to 10 by strictly following this format: "[[rating]]", for example: "Rating: [[5]]".

[Question]
{question}

[The Start of Assistant's Answer]
{answer}
[The End of Assistant's Answer]

---

## C.2 EVALUATION DETAILS

For the evaluation details, we detail the setup of the generator (i.e., PM and RLHF model) and classifier (i.e., RM), respectively.

• Reward model: the reward model is evaluated on the corresponding test split of the preference benchmark based on accuracy (i.e., whether the RM can distinguish the better and worse response in the context of the given instruction.)

• Policy and RLHF model: we follow the general principle of MT-Bench (Zheng et al., 2023). Specifically, we use their instruction (Prompt 1) to prompt GPT-4 for measuring the quality of the responses from the policy and RLHF models. GPT-4 will assign a quality score, ranging from 0 to 10, to measure the quality of the response.

## C.3 SANITY CHECK SETUP

In the sanity check for the capacity of the RM and PM, our primary objective is to verify that both models maintain comparable performance across different stages of the training process. Specifically, we aim to ensure that: (1) the RM consistently distinguishes between better and worse responses as per the instructions used in SFT and RL training; (2) the PM sustains its generation quality with instructions from the RL training dataset.

To achieve this, we utilize the Stack-Exchange dataset's three segments (SFT, Preference, RL), dividing each into train, dev, and test splits. For the RM, the data distribution is 100,000/20,000/20,000, and for the PM, it is 20,000/2,000/2,000. We prepare the dataset in a format where each instruction is paired with a corresponding high-quality answer and a lower-quality candidate, ensuring the data's compatibility for training both the RM and PM. The training configurations adhere to the setup described in Appendix C.

# D IMPLEMENTATION DETAILS OF SEAM

## D.1 PROMPT USED IN SEAM$_{\text{GPT}}$

We use GPT-4 to generate worse-quality responses in SEAM$_{\text{GPT}}$, based on the prompt detailed in Prompt 2.

---

**Prompt 2.** (Prompt used in SEAM$_{\text{GPT}}$)

[System]

Using the question and its correct answer provided below, generate 30 distinct answers that are of lower quality. Each response should include one or more of the following characteristics: factual inaccuracies, misunderstandings of the core question, irrelevant information, or grammatical errors. The answers should vary in their mistakes to cover a range of common errors seen in similar topics. Format the responses as separate paragraphs for each answer.

[Question]
{question}

[Answer]
{answer}

[The Start of Assistant's Answer]
{answer}
[The End of Assistant's Answer]

---

## D.2 CASES OF SEAM$_{\text{ADV}}$

We employed several adversarial attack strategies to challenge the integrity of the reward model (RM). Specifically, for each instruction along with its corresponding better response $r^+$ and worse response $r^-$, these adversarial attacks introduce a perturbation $\alpha$ to $r^-$. The goal is for $r^- + \alpha$ to receive a higher reward score than $r^+$, thereby compromising the RM. The attacks we utilized include GA (Wang et al., 2019), Bert-Attack (Li et al., 2020), PWWS (Ren et al., 2019), KATG (Shen et al., 2022), and TextFooler (Jin et al., 2020). However, a common limitation of these methods is that they tend to produce sentences with extremely low likelihood according to the policy model. Below, we present some examples illustrating the discrepancies between the original responses and those generated by the adversarial attacks.

## D.3 SETUP OF SEAM$_{\text{CONTRAST}}$

Using a human preference dataset, we have divided it into training, development, and testing sets. The reward model is trained on the training set and ceases training once it attains optimal performance on the development set. Subsequently, it is evaluated on the test set. Our CONTRAST INSTRUCTIONS are built upon the test set in each benchmark. We establish a similarity threshold range to ensure the retrieved instruction differs from the original one ($[0.8, 0.9]$). Only instructions falling within this similarity range are retrieved.

## D.4 HUMAN EVALUATION

Since we aim to compute the degree of match between the reward outputs and human preferences, we enlist multiple human annotators to assess the quality of responses to Stack Exchange questions. Each annotator is kept unaware of the model that generated the responses, and then they are asked to give the index of the response with better quality based on tools like search engines. Since the evaluation relates to Stack Exchange, each annotator has expertise in computer science.

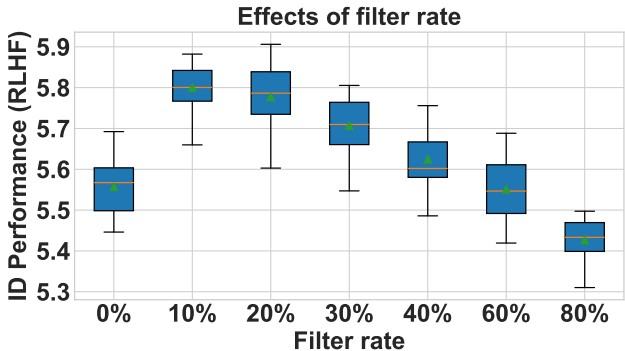

Figure 8: The effects of filter rate in RL data selection.

# E    EXTRA ANALYSIS OF LOW-SEAM DATA

## E.1    THE EFFECTS OF THE FILTERING RATE

We vary the filter rate as follows $\{10\%, 20\%, 30\%, 40\%, 60\%, 80\%\}$, and re-conduct the experiments in §7 with the rank 1 PM and RM. The results, as shown in Figure 8, demonstrate the relationship between the filter rate of data samples and the in-domain RLHF performance across various thresholds. Notably, increasing the filter rate initially enhances RLHF performance, with a peak observed at approximately 40%. Beyond this threshold, further increases in the filter rate result in a gradual decline in performance. This trend indicates an optimal range for filtering out low-seam score samples to maximize RLHF effectiveness, thereby illustrating the critical trade-off between data quantity and quality. Based on this observation, we set the filtering rate as 20%.

## E.2    THE OVERLAP RATE BETWEEN LOW-SEAM DATA ON DIFFERENT COMBINATIONS.

Following the previous setup, we examine the overlap rate of the 20% low-SEAM data across three model combinations: (1) rank 5 PM with rank 5 RM, (2) rank 3 PM with rank 3 RM, and (3) rank 1 PM with rank 1 RM. We aim to assess whether the low-SEAM data varies significantly among different model pairings. The results, illustrated in Table 3, reveal that the overlap rate between model combinations is generally high, exceeding 60%. Notably, the overlap rate increases as the differences between the models decrease.

| Model Combo | rank = 1 | rank = 3 | rank = 5 |
|-------------|----------|----------|----------|
| rank = 1    | -        |          |          |
| rank = 3    | 72%      | -        |          |
| rank = 5    | 64%      | 69%      | -        |

Table 3: The overlap rate between the 20% low-SEAM data on different model combinations, where a rank of 1 denotes using the rank 1 PM and rank 1 RM in the combination.

# F    DETAILS OF THEORETICAL ANALYSIS

In this section, we provide the results and proofs of the theoretical analysis in Section 6.

To ease notation, define

$$d_{1/\beta}(I) := D_{1/\beta}(P_r(\cdot|I; \pi^{\text{DATA}}) \| P_r(\cdot|I; \pi^{\text{SFT}})),$$
$$d_1(I) := D_{\text{KL}}(P_r(\cdot|I; \pi^{\text{DATA}}) \| P_r(\cdot|I; \pi^{\text{SFT}})),$$
$$d_\infty(I) := \sup_r \log \frac{P_r(r|I; \pi^{\text{DATA}})}{P_r(r|I; \pi^{\text{SFT}})}.$$

We first introduce the following lemma to bound $D_{\mathrm{KL}}(P_r(\cdot|I;\pi^{\mathrm{DATA}})\|P_r(\cdot|I;\pi^{\mathrm{RLHF}}))$ from below.

**Lemma 1.** *For any $0 < \beta < 1$,*

$$D_{\mathrm{KL}}(P_r(\cdot|I;\pi^{\mathrm{DATA}})\|P_r(\cdot|I;\pi^{\mathrm{RLHF}})) \geq \left(\frac{1}{\beta}-1\right)\frac{d_{1/\beta}(I)-d_1(I)}{d_{1/\beta}(I)}d_1(I).$$

Next we restate Proposition 1.

**Proposition 2** (Restatement of Proposition 1). *For any $0 < \beta < 1$ and $\eta > 0$,*

$$\mathbb{E}_{I\sim Q'_I(\cdot)}\big[D_{\mathrm{KL}}(P_r(\cdot|I;\pi^{\mathrm{DATA}})\|P_r(\cdot|I;\pi^{\mathrm{RLHF}}))\big] - \mathbb{E}_{I\sim Q'_I(\cdot)}\big[D_{\mathrm{KL}}(P_r(\cdot|I;\pi^{\mathrm{DATA}})\|P_r(\cdot|I;\tilde{\pi}^{\mathrm{RLHF}}))\big]$$

$$\geq \left\{\left(\frac{1}{\beta}-1\right)\frac{\min_{I\notin\mathcal{I}_\eta}\{d_{1/\beta}(I)-d_1(I)\}}{\max_{I\notin\mathcal{I}_\eta}d_{1/\beta}(I)}-1\right\}\cdot\eta\cdot\mathbb{P}_{I\sim Q'_I(\cdot)}(I\notin\mathcal{I}_\eta). \tag{7}$$

*Furthermore, if*

$$\mathbb{P}_{I\sim Q'_I(\cdot)}(I\notin\mathcal{I}_\eta) > 0, \ \ \max_I d_\infty(I) < \infty, \ \ \min_{I\notin\mathcal{I}_\eta}d_1(I) > 0, \tag{8}$$

*then,*

$$\liminf_{\beta\downarrow 0}\frac{\mathbb{E}_{I\sim Q'_I(\cdot)}\big[D_{\mathrm{KL}}(P_r(\cdot|I;\pi^{\mathrm{DATA}})\|P_r(\cdot|I;\pi^{\mathrm{RLHF}}))\big] - \mathbb{E}_{I\sim Q'_I(\cdot)}\big[D_{\mathrm{KL}}(P_r(\cdot|I;\pi^{\mathrm{DATA}})\|P_r(\cdot|I;\tilde{\pi}^{\mathrm{RLHF}}))\big]}{\mathbb{E}_{I\sim Q'_I(\cdot)}[D_{\mathrm{KL}}(P_r(\cdot|I;\pi^{\mathrm{DATA}})\|P_r(\cdot|I;\pi^{\mathrm{RLHF}}))]} > 0.$$

Note that since Rényi divergence of order $\alpha$ is monotonically increasing with its order (Van Erven & Harremos, 2014), the right hand side of equation 7 is always non-negative.

### F.1 PROOFS OF THEORETICAL RESULTS

*Proof of Lemma 1.* We divide the proof into 3 steps.

**Step 1.** We first prove the following under the setup in Section 6:

$$P_r(r|I;\pi^{\mathrm{RLHF}}) \propto P_r(r|I;\pi^{\mathrm{SFT}})^{1-1/\beta}P_r(r|I;\pi^{\mathrm{DATA}})^{1/\beta}. \tag{9}$$

Consider RM optimization in equation 4. From Lemma C.2 of Chen et al. (2024), the minimum value of the loss in equation 4 is achieved by

$$R^*(I,r) = Z(I) + \log\left(\frac{P_r(r|I;\pi^{\mathrm{DATA}})}{P_r(r|I;\pi^{\mathrm{SFT}})}\right), \tag{10}$$

where $Z(I)$ is any function of $I$.

From Peters & Schaal (2007); Peng et al. (2019); Korbak et al. (2022); Go et al. (2023), PM optimization problem in equation 5 has a closed-form solution $\pi^{\mathrm{RLHF}} \in \Pi$ satisfying

$$P_r(r|I;\pi^{\mathrm{RLHF}}) = \frac{1}{H(I)}P_r(r|I;\pi^{\mathrm{SFT}})\exp\big(\beta^{-1}R^*(I,r)\big),$$

where $H(I) = \int P_r(r|I;\pi^{\mathrm{SFT}})\exp\big(\beta^{-1}R^*(I,r)\big)\,\mathrm{d}r$ is a normalizing constant. Hence

$$P_r(r|I;\pi^{\mathrm{RLHF}}) = \frac{1}{H(I)}P_r(r|I;\pi^{\mathrm{SFT}})\exp\left(\frac{1}{\beta}\log\left(\frac{P_r(r|I;\pi^{\mathrm{DATA}})}{P_r(r|I;\pi^{\mathrm{SFT}})}\right) + \frac{1}{\beta}Z(I)\right)$$

$$= \frac{1}{H(I)}\exp\left(\frac{1}{\beta}Z(I)\right)P_r(r|I;\pi^{\mathrm{SFT}})^{1-1/\beta}P_r(r|I;\pi^{\mathrm{DATA}})^{1/\beta}. \tag{11}$$

This completes Step 1.

**Step 2.** In this step, we explicitly write the KL distance between $P_r(r|I; \pi^{\text{DATA}})$ and $P_r(r|I; \pi^{\text{RLHF}})$ using equation 11 in Step 1. Observe that

$$D_{\text{KL}}(P_r(\cdot|I; \pi^{\text{DATA}})\|P_r(\cdot|I; \pi^{\text{RLHF}}))$$

$$= \int P_r(r|I; \pi^{\text{DATA}}) \log \frac{P_r(r|I; \pi^{\text{DATA}})}{P_r(r|I; \pi^{\text{RLHF}})} \, \mathrm{d}r$$

$$= \int P_r(r|I; \pi^{\text{DATA}}) \log P_r(r|I; \pi^{\text{DATA}}) \, \mathrm{d}r$$

$$- \int P_r(r|I; \pi^{\text{DATA}}) \left\{ \left(1 - \frac{1}{\beta}\right) \log P_r(r|I; \pi^{\text{SFT}}) + \frac{1}{\beta} \log P_r(r|I; \pi^{\text{DATA}}) - \log H(I) + \frac{1}{\beta} Z(I) \right\} \mathrm{d}r$$

$$= \log H(I) - \frac{1}{\beta} Z(I) + \left(1 - \frac{1}{\beta}\right) D_{\text{KL}}(P_r(\cdot|I; \pi^{\text{DATA}})\|P_r(\cdot|I; \pi^{\text{SFT}})). \tag{12}$$

By definition of $H(I)$, we have

$$\log H(I) = \log \int P_r(r|I; \pi^{\text{SFT}}) \exp\left(\beta^{-1} R^*(I, r)\right) \mathrm{d}r$$

$$= \log \int \{P_r(r|I; \pi^{\text{SFT}})\}^{1-1/\beta} \{P_r(r|I; \pi^{\text{DATA}})\}^{1/\beta} \exp\left(\frac{1}{\beta} Z(I)\right) \mathrm{d}r$$

$$= \frac{1}{\beta} Z(I) + \log \int \{P_r(r|I; \pi^{\text{SFT}})\}^{1-1/\beta} \{P_r(r|I; \pi^{\text{DATA}})\}^{1/\beta} \, \mathrm{d}r \,.$$

Combined with equation 12, we have

$$D_{\text{KL}}(P_r(\cdot|I; \pi^{\text{DATA}})\|P_r(\cdot|I; \pi^{\text{RLHF}}))$$

$$= \log \int \{P_r(r|I; \pi^{\text{SFT}})\}^{1-1/\beta} \{P_r(r|I; \pi^{\text{DATA}})\}^{1/\beta} \, \mathrm{d}r + \left(1 - \frac{1}{\beta}\right) D_{\text{KL}}(P_r(\cdot|I; \pi^{\text{DATA}})\|P_r(\cdot|I; \pi^{\text{SFT}}))$$

$$= \log \int \left(\frac{P_r(r|I; \pi^{\text{DATA}})}{P_r(r|I; \pi^{\text{SFT}})}\right)^{1/\beta-1} P_r(r|I; \pi^{\text{DATA}}) \, \mathrm{d}r + \left(1 - \frac{1}{\beta}\right) D_{\text{KL}}(P_r(\cdot|I; \pi^{\text{DATA}})\|P_r(\cdot|I; \pi^{\text{SFT}}))$$

$$= \left(\frac{1}{\beta} - 1\right) \{D_{1/\beta}(P_r(\cdot|I; \pi^{\text{DATA}})\|P_r(\cdot|I; \pi^{\text{SFT}})) - D_{\text{KL}}(P_r(\cdot|I; \pi^{\text{DATA}})\|P_r(\cdot|I; \pi^{\text{SFT}}))\},$$

$$\tag{13}$$

where we used the definition of Rényi divergence. Since Rényi divergence is monotonically increasing in $\alpha$, the right hand side of equation 13 is non-negative for all $\beta > 0$.

**Step 3.** In this step, we bound $D_{\text{KL}}(P_r(\cdot|I; \pi^{\text{DATA}})\|P_r(\cdot|I; \pi^{\text{RLHF}}))$ from below. For the economy of notation, define

$$\delta(\alpha) := D_\alpha(P_r(\cdot|I; \pi^{\text{DATA}})\|P_r(\cdot|I; \pi^{\text{SFT}}))$$

for $\alpha > 1$. Since $\lim_{\alpha \downarrow 1} \delta(\alpha) = D_{\text{KL}}(P_r(\cdot|I; \pi^{\text{DATA}})\|P_r(\cdot|I; \pi^{\text{SFT}}))$, we also define $\delta(1)$ for convenience as:

$$\delta(1) := D_{\text{KL}}(P_r(\cdot|I; \pi^{\text{DATA}})\|P_r(\cdot|I; \pi^{\text{SFT}})).$$

Take any sufficiently small $\epsilon'$. From the monotonicity of Rényi divergence, we have $\delta'(\alpha) \geq 0$. This implies that

$$\log \delta(\alpha) - \log \delta(1 + \epsilon') = \int_{1+\epsilon'}^\alpha \frac{\delta'(\gamma)}{\delta(\gamma)} \mathrm{d}\gamma \geq \frac{\delta(\alpha) - \delta(1 + \epsilon')}{\delta(\alpha)}.$$

Equivalently, we have $\delta(\alpha)/\delta(1 + \epsilon') \geq \exp(1 - \delta(1 + \epsilon')/\delta(\alpha))$. Taking $\epsilon' \downarrow 0$, we have

$$\delta(\alpha) - \delta(1) \geq \delta(1) \exp\left(1 - \frac{\delta(1)}{\delta(\alpha)}\right) - \delta(1) \geq \delta(1)\left(1 - \frac{\delta(1)}{\delta(\alpha)}\right),$$

where we used $\exp(x) \geq 1 + x$ for all $x$. Combined with equation 13, we obtain

$$D_{\text{KL}}(P_r(\cdot|I; \pi^{\text{DATA}})\|P_r(\cdot|I; \pi^{\text{RLHF}})) = \left(\frac{1}{\beta} - 1\right)\{\delta(1/\beta) - \delta(1)\} \geq \left(\frac{1}{\beta} - 1\right)\frac{\delta(1/\beta) - \delta(1)}{\delta(1/\beta)}\delta(1)$$

when $1/\beta > 1$. This completes the proof. ∎

*Proof of Proposition 2.* From Lemma 1, we have

$$D_{\mathrm{KL}}(P_r(\cdot|I;\pi^{\mathrm{DATA}})\|P_r(\cdot|I;\pi^{\mathrm{RLHF}})) \geq \left(\frac{1}{\beta}-1\right)\frac{\min_{I\notin\mathcal{I}_\eta}\{d_{1/\beta}(I)-d_1(I)\}}{\max_{I\notin\mathcal{I}_\eta}d_{1/\beta}(I)}d_1(I)$$

for $I \notin \mathcal{I}_\eta$. This implies that

$$\mathbb{E}_{I\sim Q_I'(\cdot)}\big[D_{\mathrm{KL}}(P_r(\cdot|I;\pi^{\mathrm{DATA}})\|P_r(\cdot|I;\pi^{\mathrm{RLHF}}))\big] - \mathbb{E}_{I\sim Q_I'(\cdot)}\big[D_{\mathrm{KL}}(P_r(\cdot|I;\pi^{\mathrm{DATA}})\|P_r(\cdot|I;\tilde{\pi}^{\mathrm{RLHF}}))\big]$$

$$= \mathbb{E}_{I\sim Q_I'(\cdot)}\big[\{D_{\mathrm{KL}}(P_r(\cdot|I;\pi^{\mathrm{DATA}})\|P_r(\cdot|I;\pi^{\mathrm{RLHF}})) - D_{\mathrm{KL}}(P_r(\cdot|I;\pi^{\mathrm{DATA}})\|P_r(\cdot|I;\pi^{\mathrm{SFT}}))\}\mathbf{1}\{I\notin\mathcal{I}_\eta\}\big]$$

$$\geq \left\{\left(\frac{1}{\beta}-1\right)\frac{\min_{I\notin\mathcal{I}_\eta}\{d_{1/\beta}(I)-d_1(I)\}}{\max_{I\notin\mathcal{I}_\eta}d_{1/\beta}(I)}-1\right\}\mathbb{E}_{I\sim Q_I'(\cdot)}\big[D_{\mathrm{KL}}(P_r(\cdot|I;\pi^{\mathrm{DATA}})\|P_r(\cdot|I;\pi^{\mathrm{SFT}}))\mathbf{1}\{I\notin\mathcal{I}_\eta\}\big]$$

$$\geq \left(\frac{1}{\beta}-1\right)\frac{1}{\max_{I\notin\mathcal{I}_\eta}d_{1/\beta}(I)}\left\{\min_{I\notin\mathcal{I}_\eta}\{d_{1/\beta}(I)-d_1(I)\}-\frac{\beta}{1-\beta}\max_{I\notin\mathcal{I}_\eta}d_{1/\beta}(I)\right\}\eta\mathbb{P}_{I\sim Q_I'(\cdot)}(I\notin\mathcal{I}_\eta),$$

$$(14)$$

where we used the fact that $D_{\mathrm{KL}}(P_r(\cdot|I;\pi^{\mathrm{DATA}})\|P_r(\cdot|I;\pi^{\mathrm{RLHF}})) \geq \eta$ for all $I \notin \mathcal{I}_\eta$. This gives the first claim. From equation 14, with the monotonicity of Rényi divergence, we have

$$\mathbb{E}_{I\sim Q_I'(\cdot)}\big[D_{\mathrm{KL}}(P_r(\cdot|I;\pi^{\mathrm{DATA}})\|P_r(\cdot|I;\pi^{\mathrm{RLHF}}))\big] - \mathbb{E}_{I\sim Q_I'(\cdot)}\big[D_{\mathrm{KL}}(P_r(\cdot|I;\pi^{\mathrm{DATA}})\|P_r(\cdot|I;\tilde{\pi}^{\mathrm{RLHF}}))\big]$$

$$\geq \left(\frac{1}{\beta}-1\right)\frac{1}{\max_{I\notin\mathcal{I}_\eta}d_\infty(I)}\left\{\min_{I\notin\mathcal{I}_\eta}\{d_{1/\beta}(I)-d_1(I)\}-\frac{\beta}{1-\beta}\max_{I\notin\mathcal{I}_\eta}d_{1/\beta}(I)\right\}\eta\mathbb{P}_{I\sim Q_I'(\cdot)}(I\notin\mathcal{I}_\eta).$$

$$(15)$$

Again from equation 13, we have

$$D_{\mathrm{KL}}(P_r(\cdot|I;\pi^{\mathrm{DATA}})\|P_r(\cdot|I;\pi^{\mathrm{RLHF}})) \leq \left(\frac{1}{\beta}-1\right)d_{1/\beta}(I) \leq \left(\frac{1}{\beta}-1\right)\max_I d_\infty(I).$$

Therefore,

$$\frac{\mathbb{E}_{I\sim Q_I'(\cdot)}\big[D_{\mathrm{KL}}(P_r(\cdot|I;\pi^{\mathrm{DATA}})\|P_r(\cdot|I;\pi^{\mathrm{RLHF}}))\big] - \mathbb{E}_{I\sim Q_I'(\cdot)}\big[D_{\mathrm{KL}}(P_r(\cdot|I;\pi^{\mathrm{DATA}})\|P_r(\cdot|I;\tilde{\pi}^{\mathrm{RLHF}}))\big]}{D_{\mathrm{KL}}(P_r(\cdot|I;\pi^{\mathrm{DATA}})\|P_r(\cdot|I;\pi^{\mathrm{RLHF}}))}$$

$$\geq \frac{1}{\{\max_I d_\infty(I)\}^2}\left\{\min_{I\notin\mathcal{I}_\eta}\{d_{1/\beta}(I)-d_1(I)\}-\frac{\beta}{1-\beta}\max_{I\notin\mathcal{I}_\eta}d_{1/\beta}(I)\right\}\eta\mathbb{P}_{I\sim Q_I'(\cdot)}(I\notin\mathcal{I}_\eta).$$

$$(16)$$

Since $\max_I d_\infty(I) < \infty$, $\mathbb{P}_{I\sim Q_I'(\cdot)}(I\notin\mathcal{I}_\eta) > 0$, the right hand side of equation 16 is positive if

$$\min_{I\notin\mathcal{I}_\eta}\left\{d_{1/\beta}(I)-d_1(I)\right\} > \frac{\beta}{1-\beta}\max_{I\notin\mathcal{I}_\eta}d_{1/\beta}(I). \tag{17}$$

Note that for any $I \notin \mathcal{I}_\eta$, $P_r(r|I;\theta_{\mathrm{SFT}})/P_r(r|I;\theta_{\mathrm{DATA}})$ is almost surely not constant under $r \sim P_r(\cdot|I;\theta_{\mathrm{DATA}})$ since $d_1(I) \geq \min_{I\notin\mathcal{I}_\eta}d_1(I) > 0$. Thus $d_{1/\beta}(I)$ is *strictly* increasing as $\beta$ decreases. In addition, again from $\max_{I\notin\mathcal{I}_\eta}d_\infty(I) < \infty$, the right hand side of equation 17 goes to 0 as $\beta \downarrow 0$, whereas the left hand side goes to $\min_{I\notin\mathcal{I}_\eta}\{d_\infty(I)-d_1(I)\} > 0$. Therefore, the right hand side of equation 15 is strictly increasing and positive for all $\beta < \beta_0$ for some $\beta_0 \in (0,1)$. This completes the proof. ∎

# G  BROADER IMPACT

Improved Human Model Alignment: Integrating SEAM into RLHF techniques enhances the alignment between machine outputs and human values, leading to AI systems that are more ethical and responsive to user needs. This improvement is critical for increasing trust and encouraging the adoption of AI technologies across diverse sectors.

Increased Efficiency and Accessibility: Refining interactions between policy and reward models optimizes the training processes and reduces the computational resources required, making AI technologies more accessible and affordable. This democratization of AI could lead to broader innovation and application.

Misuse in Content Generation: The enhancements that improve model quality and user experience can also be exploited to create misleading information. Such misuse may pose risks of spreading misinformation and violating privacy.

```
Question : ``` < input name =" maxSalary " id =" MaxSalary " type =" number " min =" 0 " class =" form - control " ng - model =" dp .
question : ``` < input name =" maxsalary " id =" maxsalary " type =" number " min =" 0 " class =" form - control " ng - model =" dp .

maximumSalary " ng - change =" minMaxSalaryComparision ()" /> ``` this is my html input . when I add a integer value , its work properly .
maximumsalary " ng - change =" minmaxsalarycomparision ()" /> ``` this is my html input . when i add a integer value , its work properly .

but if I add 0000 ng - change only fire with first entered 0 . this is my java script . ``` $ scope . minMaxSalaryComparision = function ()
but if i add 0000 ng - change only fire with first entered 0 . this is my java script . ``` $ scope . minmaxsalarycomparision = function ()

{ alert ($ scope . dp . maximumSalary ); }; ``` any idea ? Answer : Try a ) Close all instances of CodeBlocks , then try to reopen the
{ alert ($ scope . dp . maximumsalary ); }; ``` any idea ? answer : try a ) close all instances of codeblocks , then try to reopen the

project . OR b ) Open the project file by the menu File -> Open
project . or b ) open the project file by the menu file -> clear
```

```
Question : I ' m having an issue with ` NextJS `/` webpack `. Whenever I make changes , it won ' t update anything until a full restart of
question : i ' m having an issue with ` nextjs `/` webpack `. whenever i make changes , it won ' t update anything until a full restart of

everything . This takes a really long time to compile , most recently ` 6 . 4 ` seconds for the client and ` 2 . 3 ` for the server . Often
everything . this takes a really long time to compile , most recently ` 6 . 4 ` seconds for the client and ` 2 . 3 ` for the server . often

it has taken ` 20 - 30 ` seconds in the past . ` process . env . NODE_ENV ` is undefined , ` webpack - hmr ` and ` on - demand - entries `
it has taken ` 20 - 30 ` seconds in the past . ` process . env . node_env ` is undefined , ` webpack - hmr ` and ` on - demand - entries `

both always return empty / success in the browser . My guess would be that ` Next `/` Webpack ` aren ' t able to watch the files that are
both always return empty / success in the browser . my guess would be that ` next `/` webpack ` aren ' t able to watch the files that are

being changed . My ` next ` version is ` 6 . 0 . 3 ` and I ' m working in ` Ubuntu 16 . 04 . 4 lts ` in ` WSL ` on ` Windows 10 . 0 . 17134
being changed . my ` next ` version is ` 6 . 0 . 3 ` and i ' m working in ` ubuntu 16 . 04 . 4 lts ` in ` wsl ` on ` windows 10 . 0 . 17134

`. The problem persists with a custom server , I ' m calling ` app . getRequestHandler ` once , assigning it to handler , then routing get
`. the problem persists with a custom server , i ' m calling ` app . getrequesthandler ` once , assigning it to handler , then routing get

paths (`/{ p *}`) to the handler with ` Hapi v17 . 5 . 2 `. Answer    : If you happen to find this " question and solution " and it is
paths (`/{ p *}`) to the handler with ` hapi v17 . 5 . 2 `. resolution : if you happen to find this " question and solution " and it is

still not working ... Try adding a `. env ` with ` CHOKIDAR_USEPOLLING = true ` on it , this fixed it for me , because I am not storing the
still not working ... try adding a `. env ` with ` chokidar_usepolling = true ` on it , this fixed it for me , because i am not storing the

files inside ` WSL ` but in a different disk on Windows ...
files inside ` wsl ` but in a different disk on windows ...
```

```
Question : I have the following java code : ``` DocumentBuilder db = DocumentBuilderFactory . newInstance (). newDocumentBuilder ();
question : i have the following java code : ``` documentbuilder db = documentbuilderfactory . newinstance (). newdocumentbuilder ();

Document doc = db . parse ( new File ("/ opt / myfile ")); ``` And `/ opt / myfile ` contains something like : ``` <? xml version =" 1 . 0 "
document doc = db . parse ( new file ("/ opt / myfile ")); ``` and `/ opt / myfile ` contains something like : ``` <? xml version =" 1 . 0 "

encoding =" UTF - 8 "?> <! DOCTYPE archive SYSTEM "../../ schema / xml / schema . dtd "> ... ``` I get the following error : ``` java . io .
encoding =" utf - 8 "?> <! doctype archive system "../../ schema / xml / schema . dtd "> ... ``` i get the following error : ``` java . io .

FileNotFoundException : /../ schema / xml / schema . dtd ( No such file or directory ) ``` This is a large java framework that consumes an
filenotfoundexception : /../ schema / xml / schema . dtd ( no such file or directory ) ``` this is a large java framework that consumes an

XML file produced elsewhere . I think the relative path is the problem . I don ' t think it will be acceptable to change the cwd before the
xml file produced elsewhere . i think the relative path is the problem . i don ' t think it will be acceptable to change the cwd before the

JVM starts ( the path comes from a config file that is read by the JVM itself ) and I have not found a way to change the cwd while the JVM
jvm starts ( the path comes from a config file that is read by the jvm itself ) and i have not found a way to change the cwd while the jvm

is running . How do I parse this XML file with the appropriate DTD ? Answer    : You need to use a custom [ EntityResolver ]( http ://
is running . how do i parse this xml file with the appropriate dtd ? resolution : you need to use a custom [ entityresolver ]( http ://

download . oracle . com / javase / 6 / docs / api / org / xml / sax / EntityResolver . html ) to tweak the path of the DTD so that it can be
download . oracle . com / javase / 6 / docs / api / org / xml / sax / entityresolver . html ) to tweak the path of the dtd so that it can be

found . For example : ``` db . setEntityResolver ( new EntityResolver () { @ Override public InputSource resolveEntity ( String publicId ,
found . for example : ``` db . setentityresolver ( new entityresolver () { @ override public inputsource resolveentity ( string publicid ,

String systemId ) throws SAXException , IOException { if ( systemId . contains (" schema . dtd ")) { return new InputSource ( new FileReader
string systemid ) throws saxexception , ioexception { if ( systemid . contains (" schema . dtd ")) { return new inputsource ( new filereader

("/ path / to / schema . dtd ")); } else { return null ; } } }); ``` If ` schema . dtd ` is on your classpath , you can just use `
("/ path / to / schema . dtd ")); } else { return null ; } } }); ``` if ` schema . dtd ` is on your classpath , you can just use `

getResourceAsStream ` to load it , without specifying the full path : ``` return new InputSource ( Foo . class . getResourceAsStream ("
getresourceasstream ` to load it , without specifying the full path : ``` return new inputsource ( foo . class . getresourceasstream ("

schema . dtd ")); ```
schema . dtd ")); ```
```

Figure 9: Case comparisons between the original and adversarial responses generated by text attacks. The differences are highlighted in RED.

