# OpenReview forum: "It Takes Two: On the Seamlessness between Reward and Policy Model in RLHF"
_ICLR.cc/2025/Conference — Submitted to ICLR 2025_

### Official Review · Reviewer_nYdE · 2024-10-31

**Soundness:** 2
**Presentation:** 2
**Contribution:** 2
**Rating:** 3
**Confidence:** 3

**Summary:**

The paper tries to analyze the interaction between the reward model (RM) and the trained policy model (PM), emphasizing the importance of consistency between them, named "seamlessness". It first shows that better RMs or PMs do not necessarily lead to better performances after RLHF (though it usually does), and tries to explain this because of lack of seamlessness. Then the paper motivates increasing seamlesness by removing the prompts on which the RM may be inacurrate. To this end, they introduce 3 seamlessness measures, SEAM_contrast, SEAM_GPT and SEAM_Adv, withut relying on human annotation: they can be used to filter out or augment the prompt dataset to improve RLHF performance.

**Strengths:**

* Understanding the interactions betwee the RM and the PM is an important and unexplored topic.
* The analysis showing that better RMs do not lead to better policies is interesting, though not novel.

**Weaknesses:**

1.  The proposed SEAM methods (SEAM_contrast, SEAM_GPT, SEAM_Adv) seem to contradict the paper's focus on the interaction between the RM and policy.  Instead of tailoring the RM to the PM, these methods filters out challenging examples for the RM independently of the PM. An approach that explicitly considers the interplay between the RM and PM would be more consistent with the paper's main argument.

2. Moreover, these 3 methods are computationally expensive, requiring large-scale data retrieval or GPT generations, limiting their practical applicability.

3.  The reported 40% mismatch rate appears consistent with the 66.08% accuracy of the best model in Figure 2. It would also be nice reporting the match rate of other RMs.

4. The theoretical analysis in Proposition 1 lacks clarity and depth. The proposition itself could be simplified by removing the unnecessary denominator.

5. Nit: Scaling the RM should solve most of the problems according to the literature, thus I am sceptical about the Appendix B. If you find this result consistently, I would be worth investigating.

**Questions:**

* To analyze the interactions between diverse RMs and PMs, the study could consider different architectures and sizes (rather than reducing the training dataset size) as done in "Llama 2: Open Foundation and Fine-Tuned Chat Models". Then we could try to answer the following questions "is a llama RM better for a Llama policy, or should we actually use another pretraining" etc. Here the findings might be explain by inappropriate hyperparameters, and notably by the use of LoRA hyperparmaeters that might underfit larger datasets.

* How does your figure 2 compare to the results from "How to Evaluate Reward Models for RLHF" showing that accuracy is actually the best signal for detecting the RM (despite its limitations).

* What is the vertical dimension of Figure 4 continuous ? and not discrete ?

---

### Official Review · Reviewer_mfDi · 2024-11-01

**Soundness:** 3
**Presentation:** 2
**Contribution:** 3
**Rating:** 6
**Confidence:** 3

**Summary:**

This study introduces the concept of "seamlessness" to improve Reinforcement Learning from Human Feedback (RLHF), focusing on the alignment between policy models (PMs) and reward models (RMs). The authors identify a "saturation phenomenon," where advancements in PMs and RMs don’t lead to better RLHF outcomes, partly due to a 35% mismatch rate between RM scores and human preferences. To address this, they introduce SEAM, an automatic metric to quantify PM-RM discrepancies. Experiments show that SEAM-filtered data selection and SEAM-guided model augmentation improve RLHF performance by 4.5% and 4%, respectively. This study highlights the critical alignment between PMs and RMs for effective RLHF.

**Strengths:**

- This paper tackles an important and current topic in RLHF with a method that’s both intuitive and straightforward.
- It lays out a clear research question and problem statement, supporting its claims with well-designed and diverse experiments.
- Core concepts, like the "saturation phenomenon," are explained in an accessible way that makes it easy for even new readers to follow. The use of visual aids is particularly effective, helping to deepen understanding and clarify key ideas.
- The paper offers both theoretical insights and empirical evidence for the effectiveness of its approach. Especially interesting is its effort to explain the filtering effect by analyzing the "effect of bad instruction" from a post-filtering perspective, which adds an engaging layer to the discussion.

**Weaknesses:**

- Certain parts of the writing lack clarity:
  - In Section 4, the explanation regarding Figure 3 under the sanity check section is not entirely clear. According to the Appendix, PM model quality is rated with a 1-10 score via the LLM-as-a-Judge method, but Figure 3 displays it simply as a percentage. A more detailed explanation of how the 1-10 score is converted to a percentage is needed. The same applies to RM quality.
  - Definition (1) aims to define "seamlessness." However, $S(I, R\_{\theta}, \pi^{\text{SFT}})$ is a metric that describes "seam-ness" by being proportional to RM misjudgment $\epsilon(r, R\_{\theta})$.
  - In Section 5.2, the authors describe three ways to construct samples ($\text{SEAM}\_{\text{Contrast}}$, $\text{SEAM}\_{\text{GPT}}$, $\text{SEAM}\_{\text{Adv}}$) to automate the quantification of seamlessness, which initially gives the impression of an augmentation-based approach. Although it becomes clear upon closer reading that these methods are intended to calculate the filtering criterion $\epsilon(r\_{i}, R\_{\theta})$, this initially gives the impression of an augmentation-based rather than filtering-based approach, leading to some confusion while reading.

- The placement of certain visual content is inadequate. For instance, Figure 2 is excessively large, and there is no margin between its caption and the main text, which affects readability.

- Performance evaluation across multiple datasets is insufficient. Testing on datasets beyond StackExchange is necessary.

- It is essential to verify if similar results can be reproduced with other LLMs in addition to LLaMa.

- The comparison study is lacking. Comparative evaluations with dataset filtering-based approaches, reward over-optimization methods focusing on RM and LM discrepancies, and other offline RLHF methods are necessary.

**Questions:**

- In the sanity check experiment in Section 4, the authors rely on the LLM-as-a-Judge method to measure the quality of the PM model. Could this evaluation approach itself be flawed?
- In Figure 4 of Section 4.2, does Response B represent a low-quality response generated by the PM model (rank 5)? If so, why would human annotators prefer Response B despite its low quality? Doesn’t this imply that $\mathcal{Q}\_{PM}$ may be an unreliable metric?
- Does $\pi^{\text{DATA}}$ refer to $\mathcal{D}\_{rl}$, which includes bad samples such as $\text{SEAM}\_{\text{Contrast}}$, $\text{SEAM}\_{\text{GPT}}$, or $\text{SEAM}\_{\text{Adv}}$?

---

### Official Review · Reviewer_sNAF · 2024-11-04

**Soundness:** 3
**Presentation:** 3
**Contribution:** 3
**Rating:** 5
**Confidence:** 4

**Summary:**

This paper introduces the concept of seamlessness to explain the saturation performance phenomenon of the policy model during RLHF. It proposes the SEAM metric to measure the performance difference between the policy model and the reward model. The motivation behind this work is clear and meaningful. However, the paper lacks deeper analysis of the experimental phenomena and the experiments are not sufficiently comprehensive.

**Strengths:**

1. The concept of seamlessness is a novel approach to understanding the saturation performance in policy models during RLHF.

2. The introduction of the SEAM metric provides a new way to measure the performance gap between the policy model and the reward model.

3. The motivation for this work is clear and addresses an important issue in the field.

**Weaknesses:**

1. How does the SEAM metric change during the alignment training process? Does the difference between the policy model and the reward model increase gradually with training?

2. Could filtering out data using the SEAM metric potentially reduce the diversity of the training data? For instance, could this lead to the model ignoring difficult data points?

3. The experimental section should include more models and datasets, such as Qwen and Mistral, and evaluate performance in helpful and harmless scenarios to enhance the credibility of the results.

4. Can this method guide the optimization of the reward model by helping to filter training data and optimize the annotation process?

**Questions:**

please see weaknesses

---

### Meta-Review · Area_Chair_h9r1 · 2024-12-18

**Metareview:**

This paper introduces the concept of "seamlessness" to analyze the interaction between policy models and reward models in RLHF. It proposes an automatic metric to quantify policy-model and reward-model discrepancies, and demonstrates its application in filtering training data and augmenting models. Experimental results claim improvements of 4.5% and 4% in RLHF performance using SEAM-guided methods.

While the interaction between the policy model and the reward model during RLHF is interesting, the proposed method focus on filtering RM-challenging data rather than fostering PM-RM interplay, contradicting the central thesis of the paper. Experiments lack breadth, with insufficient evaluation across diverse datasets, models, and practical scenarios. Writing is unclear in several sections.

**Additional Comments On Reviewer Discussion:**

Authors did not respond to the concerns raised by reviewers.

---

### Decision · Program_Chairs · 2025-01-22

Reject